# The mediation effect of general self-efficacy in relation to procrastination and sense of coherence among adults with attention deficit hyperactivity disorder

Agnieszka Malinowska*, Wojciech Rodzeń*

Department of Social Sciences, Institute of Psychology, University of Szczecin, Szczecin, Poland

* agnieszka.malinowska@usz.edu.pl (AM); wojciech.rodzen@usz.edu.pl (WR)

## Abstract

Procrastination, defined as a tendency to delay starting or finishing tasks despite awareness of negative consequences, is a common problem among adults diagnosed with Attention Deficit Hyperactivity Disorder (ADHD). In the context of ADHD, procrastination not only exacerbates difficulties related to concentration and organization but also affects overall life functioning, interpersonal relationships, and mental health. Therefore, identifying protective factors that can mitigate the impact of procrastination on the lives of people with ADHD is crucial. The aim of this study was to underscore the importance of general self-efficacy as a potential intervention target for improving the psychological well-being in terms of the sense of coherence. The study involved 180 people aged 18–56 years. All the study participants had been diagnosed with ADHD, some of whom were undergoing therapy and pharmacological treatment. The participants filled out a self-report questionnaire consisting of demographic variable measurement and three psychometric tools. The results of the analyses indicated a negative relationship between procrastination and sense of coherence, as well as between procrastination and self-efficacy, and a positive relationship between self-efficacy and sense of coherence. The obtained results of the mediation analysis support the research hypothesis of a partial mediation effect that a self-efficacy weakens the relationship between procrastination and a sense of coherence. The study contributes to the understanding of ADHD in adults and highlights the critical role of psychological factors in managing its symptoms and improving quality of life.

## Introduction

Attention deficit hyperactivity disorder (ADHD) is a neurodevelopmental condition characterised by symptoms of inattention, hyperactivity, and impulsivity [1,2]. Once considered a childhood disorder, ADHD is now recognised as persisting into

**Data availability statement:** The data that support the findings of this study are publicly available in the Open Science Framework (OSF) repository at the following link: https://osf.io/9sbze/.

**Funding:** The author(s) received no specific funding for this work.

**Competing interests:** The authors have declared that no competing interests exist.

adulthood, affecting approximately 2.5% of adults and 8.4% of children [3]. Older studies often estimate the prevalence of ADHD at 1–20% [4] or 3–9% [5]. ADHD is diagnosed twice as often in boys as in girls, whereas in adults these proportions change [6]. Although expression of ADHD symptoms may vary with age and context, core symptoms remain consistent and often impair occupational, social, and emotional functioning [6,7].

According to the Diagnostic and Statistical Manual of Mental Disorders (DSM-5) [8], ADHD symptoms are grouped into two domains: inattention and hyperactivity/impulsivity, and three subtypes of ADHD are distinguished: combined, predominantly inattentive and predominantly hyperactive-impulsive. The International Classification of Diseases (ICD-11) [9] uses the same terminology. Despite historical discussions on environmental influences, ADHD is currently viewed as a disorder with a neurobiological basis, with social factors acting primarily as mediating factors of symptom expression [10–12].

One of the behavioural manifestations often associated with ADHD is procrastination, understood as the voluntary delay of intended tasks despite expecting negative consequences [13]. Procrastination can be driven by both cognitive and motivational deficits. Passive procrastinators tend to avoid tasks due to self-doubt and low confidence, whereas active procrastinators delay strategically to work under time pressure [14]. In individuals with ADHD, procrastination is linked to inattention symptoms [15] and difficulties in prospective memory [16], suggesting a connection between executive dysfunction and self-regulatory failure. Furthermore, a negative correlation is observed between procrastination and a sense of coherence [17], as well as between procrastination and a self-efficacy [18].

The sense of coherence (SOC), proposed by Antonovsky [19], reflects a global orientation expressing the extent to which one perceives life as comprehensible, manageable, and meaningful [20,21]. It serves as a protective factor facilitating effective coping with stress and maintaining psychological well-being. Individuals with a stronger SOC experience fewer emotional and behavioural problems related to ADHD symptoms [22]. SOC develops through consistent life experiences that enhance predictability and resourcefulness [23]. The sense of coherence is a quality that acts as a protective factor against the effects of ADHD symptoms. Individuals with a stronger sense of coherence reported a lower degree of difficulties experienced in the area of emotional distress, antisocial behaviours or substance abuse compared to those with a lower sense of coherence [22].

Closely related to SOC is self-efficacy, defined by Bandura [24] as one's belief in their capacity to execute actions necessary to achieve specific goals. High self-efficacy fosters persistence, goal commitment, and motivation, and it contributes to adaptive coping and better health outcomes [25,26]. While SOC represents a broader, relatively stable orientation toward life, self-efficacy reflects task-specific confidence and motivational readiness to act [27,28]. Importantly, self-efficacy and SOC are positively correlated [29].

Given that procrastination among adults with ADHD may arise from self-regulatory and motivational deficits, both SOC and self-efficacy may function as important

psychological resources. Previous studies indicate that low self-efficacy mediates the relationship between ADHD symptoms and procrastination [18], yet the interplay between self-efficacy, SOC, and procrastination within this population remains understudied.

## Purpose of the study

The present study aims to examine the relationships between procrastination, sense of coherence, and self-efficacy in adults with ADHD, and to verify whether general self-efficacy mediates the association between procrastination and sense of coherence. The obtained results will provide information on the possibility of implementing effective therapeutic interventions for adults diagnosed with ADHD.

### *H1. A self-efficacy is a mediator in the relationship between procrastination and a sense of coherence.*

Analysis of existing studies suggests a significant negative association between procrastination and self-efficacy [18], as well as between procrastination and sense of coherence [17]. The literature also indicates a positive relationship between self-efficacy and sense of coherence [29]. However, empirical evidence verifying the mediating role of self-efficacy in the relationship between procrastination and sense of coherence remains limited, which prompted us to explore this hypothesis.

## Matherials and methods

### Study design

In the study, the convenience sampling method was used. Participants were recruited through online announcements posted in closed ADHD-focused support groups and forums on social media (e.g., Facebook communities dedicated to ADHD in adults). This approach allowed us to reach individuals with a confirmed ADHD diagnosis who may not necessarily be under current clinical care but are engaged in ADHD-related communities. At the beginning of the survey, all respondents were asked to indicate whether they had ever been diagnosed with ADHD by answering the following question: "Have you ever been diagnosed with ADHD?" with three possible response options: (1)*Yes, by a specialist*, (2)*No, but I suspect I have ADHD*, (3)*No, and I do not suspect I have ADHD*. Only individuals who selected the first option could participate in the study and they were included in the analyses. The ADHD diagnosis was therefore self-reported and not independently verified with standardized clinical tools or documentation. This procedure may involve a risk of misclassification, which is acknowledged as a study limitation. Respondents completed an online questionnaire battery hosted on a secure survey platform. All participants were adults (aged 18 or older) and provided electronic informed consent after receiving detailed information about the study purpose, procedures, confidentiality, and voluntary participation. The recruitment period for the study lasted from April 21, 2023, to March 26, 2024. The Ethics Committee for Research Projects of the Institute of Psychology at the University of Szczecin reviewed and approved both the study design and the consent procedure (approval no. KB 11/2023).

### Data measurement

To measure procrastination, the *Pure Procrastination Scale* (PPS) [13] was used in the Polish adaptation by Stępień and Cieciuch[Unpublished]. The participants respond to 12 questions, rating themselves on a 5-point scale where 1 indicates *it does not describe me at all* and 5 indicates *it describes me really accurately*. The tool measures the general level of procrastination as well as its three dimensions: decisional, behavioural and maladaptive. In the study, satisfactory Cronbach's alpha reliability coefficients were obtained for the overall score (0.88) and for the three dimensions of procrastination (0.82–0.87).

The sense of coherence was measured with the *SOC-29 Life Orientation Questionnaire* [30] in Antonovsky's original version [37]. The tool assesses the overall level of sense of coherence as well as its three dimensions: comprehensibility,

manageability and meaningfulness through 29 questions, to which the subjects respond on a 7-point scale. Higher scores signify a higher sense of coherence and its components. The tool was characterised by a high level of reliability, both for the overall result of sense of coherence ($\alpha = 0.88$) and the sense of comprehensibility ($\alpha = 0.70$), manageability ($\alpha = 0.77$) and meaningfulness ($\alpha = 0.81$).

The measurement of a self-efficacy was carried out using the *General Self-Efficacy Scale* (GSES) by Schwarzer [31], with the Polish adaptation by Juczyński [32]. The single-factor tool includes 10 questions, and the respondents are tasked with answering them on a 4-point scale ranging from 1 – *no*, to 4 – *yes*. A higher score indicates a higher self-efficacy. The tool was characterised by a high degree of reliability ($\alpha = 0.88$).

## Statistical methods

All analyses were performed using *IBM SPSS Statistics 26*. To verify the hypothesis regarding the mediating effect of a self-efficacy between procrastination and a sense of coherence, *PROCESS 4.2* macro by A. F. Hayes [33] was used. The mediation model analysis was conducted using a bootstrapping method with a random draw of 5000 samples.

Since the analysis of mediating effects is based on regression analysis, the initial stage of calculations involved checking the degree to which the assumptions of linearity of the model, homoscedasticity, normal distribution of random components and collinearity were met. Additionally, the data has been checked for the presence of influential outliers. To achieve this goal, analyses were conducted using Cook's and Mahalanobis distance measures as well as influence values. The criterion used to reject a particular observation was identified as the failure to meet two out of three threshold values. Based on the preliminary analyses, it has been determined that the model under examination meets all the criteria appropriate for the theoretical assumptions of regression analysis, and the data itself has no missing values or outliers.

## Results

### Participants and descriptive data

The study involved 180 participants aged between 18–56 years ($M = 29.58$; $SD = 6.86$). Most of respondents were women ($n = 157$; 87.22%), individuals with higher education ($n = 115$; 63.89%), residents of cities with over 500,000 inhabitants ($n = 95$; 52.78%) and employed persons ($n = 97$; 53.89%). All the study participants had been diagnosed with ADHD. Furthermore, 117 individuals (65%) declared that they have participated in therapy at some point, and 114 individuals (63.33%) are undergoing pharmacotherapy in connection with an ADHD diagnosis in adulthood. Most of respondents received an ADHD diagnosis as adults ($n = 162$; 90%), 6 individuals were diagnosed with ADHD during adolescence (3.33%), and 12 people were diagnosed with ADHD in childhood – up to 12 years of age (6.67%). Detailed characteristics of the study sample are presented in Table 1.

### Main results

The calculations that preceded the analysis of the mediation model involved verifying the strength and directions of interdependence between individual variables based on the obtained values of Pearson's r correlation coefficients (Table 2).

The correlation results obtained indicate statistically significant ($p < 0.001$) relationships between the overall scores of procrastination, self-efficacy and sense of coherence, as well as all factors of procrastination and sense of coherence. Based on the analysis of inter-variable dependencies included in the final mediation model, it was demonstrated that procrastination correlates moderately and negatively with a self-efficacy ($r = -0.47$; $p < 0.001$) and sense of coherence ($r = -0.58$; $p < 0.001$), while sense of coherence correlates strongly and positively with a self-efficacy ($r = 0.66$; $p < 0.001$).

The analysis of the mediation effect of a self-efficacy on the relationship between procrastination and sense of coherence (Fig 1) showed that the model was well-fitted to the data ($F(2,177) = 101.13$; $p < 0.001$) and explained approximately 53% of the variance in the dependent variable ($_{adj}R^2 = 0.53$). In accordance with the classic assumptions of mediation

**Table 1. Sociodemographic and clinical characteristics of the sample.**

| Variable/category | n | % |
| --- | --- | --- |
| **Sex** | | |
| Female | 157 | 87.2 |
| Male | 23 | 12.8 |
| **Level of education** | | |
| Primary | 1 | 0.5 |
| Lower secondary | 1 | 0.5 |
| Basic vocational | 3 | 1.7 |
| Secondary | 60 | 33.3 |
| Higher | 115 | 63.9 |
| **Place of residence (size of locality)** | | |
| Rural area | 23 | 12.8 |
| Town up to 50,000 inhabitants | 25 | 13.9 |
| Town from 50,000–150,000 inhabitants | 8 | 4.4 |
| City from 150,000–500,000 inhabitants | 29 | 16.1 |
| City over 500,000 inhabitants | 95 | 52.8 |
| **Occupational status** | | |
| Neither studying nor employed | 14 | 7.8 |
| Studying | 18 | 10 |
| Employed | 97 | 53.9 |
| Studying and employed | 51 | 28.3 |
| **Pharmacotherapy related to ADHD** | | |
| Yes | 114 | 63.3 |
| No | 66 | 36.7 |
| **Participation in therapy related to ADHD in adulthood** | | |
| Yes | 63 | 35 |
| No | 117 | 65 |
| **Time of ADHD diagnosis** | | |
| In childhood (up to age 12) | 12 | 6.7 |
| In adolescence (ages 13–18) | 6 | 3.3 |
| After age 18 | 162 | 90 |
| **Consultation with a psychologist related to ADHD in childhood[a]** | | |
| Yes | 7 | 38.9 |
| No | 11 | 61.1 |
| **Use of therapeutic strategies by parents/teachers in childhood to support coping with ADHD[a]** | | |
| Yes | 6 | 33.3 |
| No | 12 | 66.7 |

[a]Responses excluding respondents diagnosed with ADHD after the age of 18

analysis [34,35], statistically significant path coefficients were obtained between the explanatory variable – procrastination and the mediating variable – a self-efficacy (β = −0.48; $p < 0.001$), as well as between the mediating variable and the outcome variable – a sense of coherence (β = 0.50; $p < 0.001$). Furthermore, a statistically significant and negative direct effect of procrastination and sense of coherence was identified (β = −0.58; $p < 0.001$), which was weaker when considering the mediating role of a self-efficacy obtaining indirect effect (β = −0.35; $p < 0.001$). The obtained result of the indirect effect (*Indirect* = −0.48; 95%*CI*[−0.66;-0.33]) indicates that the model demonstrates a statistically significant partial mediation

**Table 2. The values of Pearson's r correlation coefficients between individual overall results of psychological variables and their factors.**

|  | (1) | (2) | (3) | (4) | (5) | (6) | (7) | (8) |
|---|---|---|---|---|---|---|---|---|
| **Sense of coherence (1)** | – |  |  |  |  |  |  |  |
| Comprehensibility (2) | 0.82*** | – |  |  |  |  |  |  |
| Manageability (3) | 0.90*** | 0.64*** | – |  |  |  |  |  |
| Meaningfulness (4) | 0.77*** | 0.35*** | 0.62*** | – |  |  |  |  |
| **Procrastination (5)** | −0.58*** | −0.45*** | −0.46*** | −0.55*** | – |  |  |  |
| Decisional (6) | −0.54*** | −0.39*** | −0.43*** | −0.54*** | 0.84*** | – |  |  |
| Behavioral (7) | −0.55*** | −0.43*** | −0.43*** | −0.52*** | 0.95*** | 0.72*** | – |  |
| Nonadaptive (8) | −0.37*** | −0.28*** | −0.31*** | −0.33*** | 0.71*** | 0.42*** | 0.55*** | – |
| **Self-efficacy** | 0.66*** | 0.46*** | 0.63*** | 0.57*** | −0.47*** | −0.39*** | −0.44*** | −0.35*** |

*** $p < 0.001$.

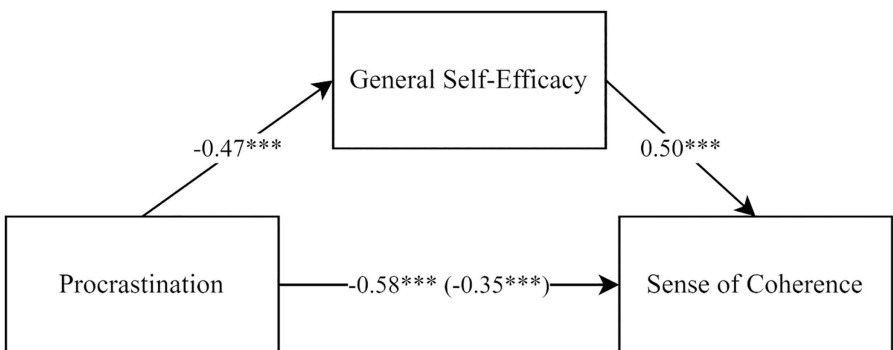

**Fig 1. Results of the mediation analysis (standardized weights).** *** $p < 0.001$.

effect. In addition to the direct and indirect effects, we estimated the proportion of the total effect mediated by self-efficacy. The indirect effect accounted for approximately 40% of the overall association between procrastination and sense of coherence, indicating a substantial partial mediation.

## Discussion

The primary objective of this study was to examine whether a self-efficacy mediates the relationship between procrastination and a sense of coherence in adults with ADHD. It has already been proven that a sense of coherence is a protective resource in general [19,20], as well as among adults diagnosed with psychomotor overactivity with attention deficit [22]. Procrastination, on the other hand, can be seen as an integral part of the functioning of adults with the aforementioned diagnosis [36], particularly in the context of attention deficit as a symptom of ADHD [15]. A negative relationship is observed between procrastination and the sense of coherence [17,37]. This means that the tendency to procrastinate, despite being aware of the negative consequences of such behaviour, co-occurs with a lower overall sense of coherence, which is an individual's ability to perceive incoming stimuli as meaningful, worthy of engagement and organised [20]. In light of the above, consideration was given to whether a self-efficacy could change this relationship.

The results of our study suggest that a self-efficacy mediates the relationship between procrastination and a sense of coherence among adults diagnosed with ADHD. The research hypothesis has been confirmed. This means that the procrastination of various activities and decision-making experienced by individuals with ADHD explains their reduced ability

to understand the world and stimuli, as well as to interpret them as worthy of engagement in coping with them, along with a lower sense of being able to deal with difficulties. For patients with ADHD, this may lead to more far-reaching consequences such as antisocial behaviours, abuse of psychoactive substances or broadly understood emotional suffering [22], due to the lack of a strong protective resource. This result is an indication for clinicians who are managing the therapy of individuals diagnosed with ADHD. It is worthwhile to incorporate cognitive-behavioural interventions that strengthen a self-efficacy into therapy, as they weaken the negative connection between procrastination, as an integral part of the disorder, and the sense of coherence, which is shaped by a multitude of experiences and direct addressing it in therapy may be challenging.

In further research, it would be valuable to consider participation in therapy as a moderator of the relationship between procrastination and a self-efficacy, as well as a moderator of the relationship between a self-efficacy and a sense of coherence. This would confirm the previously mentioned thesis that enhancing a self-efficacy is an important therapeutic intervention for individuals with ADHD. Another contributing aspect could be an experimental study in which a self-efficacy is directly enhanced through therapy, in order to eventually compare the outcomes of the described variables with a group of individuals whose self-efficacy was not directly reinforced.

## Implications and limitations

This study has several methodological limitations. Participants were recruited online from ADHD-focused support groups and social media forums, which may have introduced self-selection bias and limited the representativeness of the sample. The ADHD diagnosis was self-reported – participants indicated whether a specialist had ever diagnosed them, but no independent verification through clinical assessment or documentation was performed. Consequently, potential diagnostic misclassification cannot be excluded.

Moreover, the online recruitment strategy carries an inherent self-selection bias. Individuals active in ADHD-related online communities may represent a subset of the broader ADHD population who are more aware of their condition, more motivated to participate in psychological research, or differ in clinical or sociodemographic characteristics. Consequently, the sample may not be representative of all adults with ADHD, and the findings should be interpreted with caution regarding their generalizability. Cultural and geographical factors related to the composition of online support groups may have also influenced the results.

Another limitation concerns the electronic form of the research, which implies a lack of control over the individuals participating in the study. Conducting a cross-sectional study limits the interpretation of the obtained results over time. The application of longitudinal study would enable the identification of certain patterns in the changes in functioning of individuals with ADHD, as well as a deeper understanding of the causal relationships between various variables. Furthermore, random selection of the sample would broaden the scope of the obtained results to the general population. It would probably be more reliable to check whether a self-efficacy changes the relationship between procrastination and a sense of coherence in the conditions of a stationary study.

Another limitation concerns treatment status. A significant proportion of participants reported undergoing psychotherapy or pharmacological treatment for ADHD. As previous research has shown that pharmacological treatment can influence personality traits and behavioral functioning in ADHD [38], it is possible that treatment status may have affected the relationships between procrastination, self-efficacy, and sense of coherence observed in our study. However, we did not control for treatment status in our analyses, which may have introduced additional variance. Future research should examine whether therapeutic or pharmacological interventions moderate the relationships among these psychological variables.

An additional limitation is the lack of data on psychiatric comorbidities. ADHD in adults is frequently comorbid with other mental health disorders, including depression [39], bipolar disorder [40], and substance use disorders [41]. Because we did not assess these conditions in our sample, we cannot exclude the possibility that comorbidities influenced the relationships observed between procrastination, self-efficacy, and sense of coherence.

Despite these limitations, the present study provides insights into the psychological mechanisms related to procrastination in adults with ADHD. The outcomes of the study in question suggest that a critical factor for maintaining a paramount resource such as the sense of coherence at a slightly higher level, irrespective of procrastination, which appears to be a symptomatic factor among individuals with ADHD, would be the enhancement of a self-efficacy.

## Author contributions

**Conceptualization:** Agnieszka Malinowska.

**Data curation:** Agnieszka Malinowska, Wojciech Rodzeń.

**Formal analysis:** Wojciech Rodzeń.

**Methodology:** Wojciech Rodzeń.

**Project administration:** Agnieszka Malinowska.

**Software:** Wojciech Rodzeń.

**Supervision:** Agnieszka Malinowska.

**Validation:** Wojciech Rodzeń.

**Visualization:** Agnieszka Malinowska.

**Writing – original draft:** Agnieszka Malinowska.

**Writing – review & editing:** Wojciech Rodzeń.

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
