## [Decision Letter · Decision Letter 0]

30 Jul 2025

Dear Dr. Rodzeń,

Thank you for submitting your manuscript to PLOS ONE. After careful consideration, we feel that it has merit but does not fully meet PLOS ONE’s publication criteria as it currently stands. Therefore, we invite you to submit a revised version of the manuscript that addresses the points raised during the review process.

We look forward to receiving your revised manuscript.

Kind regards,

Nicholas Aderinto Oluwaseyi

Academic Editor

PLOS ONE

Journal Requirements:

2. Please describe in your methods section how capacity to provide consent was determined for the participants in this study. Please also state whether your ethics committee or IRB approved this consent procedure. If you did not assess capacity to consent please briefly outline why this was not necessary in this case.

3. In the online submission form you indicate that your data is not available for proprietary reasons and have provided a contact point for accessing this data. Please note that your current contact point is a co-author on this manuscript. According to our Data Policy, the contact point must not be an author on the manuscript and must be an institutional contact, ideally not an individual. Please revise your data statement to a non-author institutional point of contact, such as a data access or ethics committee, and send this to us via return email. Please also include contact information for the third party organization, and please include the full citation of where the data can be found.

4. We note that you have referenced (Stępień M, Cieciuch J et al. [47]) which has currently not yet been accepted for publication. Please remove this from your References and amend this to state in the body of your manuscript: (Stępień M, Cieciuch J et al. [Unpublished]”) as detailed online in our guide for authors

http://journals.plos.org/plosone/s/submission-guidelines#loc-reference-style .

5. Please ensure that you refer to Figure 1 in your text as, if accepted, production will need this reference to link the reader to the figure.

6. Please remove your figures from within your manuscript file, leaving only the individual TIFF/EPS image files, uploaded separately. These will be automatically included in the reviewers’ PDF.

Reviewers' comments:

Reviewer's Responses to Questions

**Comments to the Author**

1. Is the manuscript technically sound, and do the data support the conclusions?

Reviewer #1: No

2. Has the statistical analysis been performed appropriately and rigorously?

Reviewer #1: No

3. Have the authors made all data underlying the findings in their manuscript fully available?

Reviewer #1: Yes

4. Is the manuscript presented in an intelligible fashion and written in standard English?

Reviewer #1: Yes

Reviewer #1: This is an interesting article that investigates the relationship between self-efficacy, procrastination, and sense of coherence. The topic is relevant and worth further exploration. However, major revisions are required to improve the overall quality and clarity of the manuscript:

1. The Introduction section should not be divided into subparagraphs. Instead, it should be structured in a coherent, narrative form. Moreover, it should be significantly shortened and focused specifically on the rationale and background relevant to the current study.

2. The description of data analysis is missing from the Methods section and is incorrectly placed in the Results section. This needs to be corrected.

3. The manuscript states that “the convenience sampling method was employed”. This should be better explained and contextualized. Moreover, the authors should clarify the clinical setting in which participants were selected and recruited.

4. The number of participants and sample description should be reported in the Results section, not in the Methods. The authors are strongly encouraged to follow the STROBE guidelines (von Elm et al., 2007; doi: 10.1016/S0140-6736(07)61602-X) for reporting observational studies.

5. A table reporting sociodemographic and clinical characteristics of the sample should be included in the Results section of the manuscript.

6. Traditional mediation analyses report both direct and indirect effects to estimate the proportion of the association that is mediated. This information is missing and should be added.

7. Although personality traits and behaviors in ADHD can change following pharmacological treatment (Takım et al., 2024; doi: 10.5152/alphapsychiatry), the manuscript does not address treatment status. This should be included as an additional limitation of the study.

8. ADHD in adults is frequently comorbid with other psychiatric disorders, such as depression (Fu et al., 2025; doi: 10.3389/fpsyt.2025.1597559), bipolar disorder (Bartoli et al., 2023; doi: 10.1177/00048674221106669), and substance use disorders (Di Nicola et al., 2024; doi: 10.1016/j.jad.2024.03.059). However, no information on these co-occurring conditions were reported, altough the potential influence of such comorbidities on the psychological variables under study. This issue should be acknowledged and discussed in the Discussion section as a limitation.

9. Other minor changes:

- The phrase “sense of self-efficacy” is redundant. “Self-efficacy” alone is sufficient and should be used throughout the manuscript.

- The authors refer to "DSM-V", which is incorrect. The correct terminology is DSM-5 and should be used consistently.

- After the first mention of Attention-Deficit/Hyperactivity Disorder, the abbreviation ADHD should be introduced and consistently used throughout the manuscript.

**Do you want your identity to be public for this peer review?** For information about this choice, including consent withdrawal, please see our Privacy Policy

Reviewer #1: No

---

## [Author Response · Author response to Decision Letter 1]

30 Aug 2025

Dear Reviewer,

We would like to express our gratitude for the time and effort you dedicated to reviewing our manuscript. Your insightful and constructive comments greatly helped us to improve the clarity, structure, and overall quality of the paper. We have revised the manuscript accordingly, and below we provide a detailed, point-by-point response to each of your comments, including a description of the changes introduced:

1. We revised the Introduction to merge subparagraphs into a coherent narrative and shortened it by removing tangential historical and neurobiological details, focusing instead on the rationale for the present study.

2. We have moved the description of data analysis from the Results section to the Matherials and Methods section.

3. We expanded the description to explain the rationale for using convenience sampling and clarified recruitment via closed ADHD-focused online communities, which included adults with a prior clinical diagnosis of ADHD.

4. We moved participant characteristics from Matherials and Methods to Results, and added a Table 1 reporting sociodemographic and clinical characteristics of the sample.

5. We have added direct, indirect, and total effects, and calculated the proportion mediated.

6. We have added a limitation acknowledging that a substantial proportion of participants were undergoing therapy or pharmacological treatment, and that treatment status may influence psychological functioning. We also addressed a ADHD comorbidities in the Limitations section.

7. We carefully revised the manuscript to ensure consistent use of “self-efficacy” (instead of “sense of self-efficacy”), corrected DSM terminology to DSM-5, and consistently introduced ADHD as the abbreviation after first mention.

---

## [Decision Letter · Decision Letter 1]

5 Oct 2025

Dear Dr. Rodzeń,

We look forward to receiving your revised manuscript.

Kind regards,

Nicholas Aderinto Oluwaseyi

Academic Editor

PLOS ONE

Journal Requirements:

Reviewers' comments:

Reviewer's Responses to Questions

**Comments to the Author**

Reviewer #1: (No Response)

2. Is the manuscript technically sound, and do the data support the conclusions?

Reviewer #1: (No Response)

3. Has the statistical analysis been performed appropriately and rigorously?

Reviewer #1: (No Response)

4. Have the authors made all data underlying the findings in their manuscript fully available?

Reviewer #1: (No Response)

5. Is the manuscript presented in an intelligible fashion and written in standard English?

Reviewer #1: (No Response)

Reviewer #1: I thank the authors for having revised the manuscript following my previous comments. While the manuscript has improved, some additional revisions are needed:

• As previously suggested, the Introduction should be significantly shortened, as it is disproportionately long (over 1500 words) especially if compared with the brevity of the Discussion section. The section on the nosological view according to DSM-5 and ICD-10 can be markedly reduced. Background information on brain functioning and neurotransmission appears unrelated to the study aims and can be removed. Theoretical considerations on psychological variables should also be shortened and presented in a more focused way.

• I recommend integrating two subsections (“Purpose of the study” and “Hypotheses”) within the Introduction paragraph.

• The recruitment process requires clarification. The authors stated that participants were recruited online and that the ADHD diagnosis was confirmed by a qualified specialist. However, it is unclear how the diagnosis was confirmed. This is important since if all diagnoses were self-reported but not independently verified with standardized clinical tools, this may introduce misclassification. This should be clearly reported in the Methods section. Moreover, in the Results section, the authors should specify how many were confirmed with ADHD and how many were without ADHD after assessment. A flowchart illustrating the selection process would improve clarity.

• In the Discussion section, the authors reported: “The primary objective of this study was to examine whether self-efficacy alters the relationship between procrastination and a sense of coherence in adults with ADHD”. I suggest replacing “alters” with “mediates” to clarify the relationship being analysed.

• The sentence “The studies presented in the article suggest that self-efficacy mediates the relationship between procrastination and a sense of coherence among adults diagnosed with ADHD” is ambiguous. Are the authors referring to their own results (“the results of our study suggest…”) or to findings from other studies? If it refers to the present results, it should be rephrased accordingly. If it refers to other research, the relevant studies must be cited. Moreover, this statement would contradict what is stated in the Introduction, where the authors noted the absence of empirical verification regarding the mediating role of self-efficacy.

• The authors reported that “A limitation of the study is the electronic form of the research, which implies a lack of control over the individuals participating in the study”. However, this part should be expanded. Indeed, the recruitment strategy relying on online ADHD-focused support groups has several methodological limitations that should be acknowledged. First, self-selection bias may have occurred, as only individuals motivated to engage in ADHD-related communities were reached, limiting the representativeness of the sample. Second, individuals active in online groups may differ from the broader ADHD population in terms of sociodemographic or clinical characteristics, reducing generalizability. Finally, cultural or geographical biases may exist depending on the composition of the online communities. All these aspects should be discussed in the Limitations section.

**Do you want your identity to be public for this peer review?** For information about this choice, including consent withdrawal, please see our Privacy Policy

Reviewer #1: No

---

## [Author Response · Author response to Decision Letter 2]

20 Oct 2025

We would like to express our sincere gratitude for the thorough and insightful feedback! Your comments were extremely helpful in improving the clarity, methodological transparency, and overall quality of our manuscript. All suggested changes have been implemented in the revised version of the paper (highlighted in red). Below, we provide a detailed summary of how each point has been addressed:

1. The "Introduction" section was substantially shortened (reduced to 500 words). We removed detailed nosological descriptions of ADHD in DSM-5 and ICD-10, as well as information on brain structures, neurotransmission, and historical context. The theoretical background on procrastination, sense of coherence, and self-efficacy was condensed and focused strictly on variables relevant to the study aims. Consequently, the numbering of references was updated to reflect these changes.

2. The subsections "Purpose of the Study" and "Hypotheses" were merged into the main body of the "Introduction" for greater conciseness and smoother narrative flow. The study aim and hypothesis are now presented in one coherent paragraph.

3. We have clarified the recruitment and diagnostic procedure in the "Materials and Methods" section. ADHD diagnosis was self-reported, based on participants’ responses. Only individuals who selected “Yes, by a specialist” could proceed to the survey and were included in the dataset. Participants who chose other responses were automatically excluded from participation at the qualification stage, so no data were collected from them. Therefore, there were no additional exclusion steps after data collection, and a flowchart was not applicable in this case. We have also explicitly acknowledged in the "Limitations" section that the diagnosis was self-reported and not independently verified with clinical tools or documentation, which may introduce potential misclassification bias.

4. The term “alters” was replaced with “mediates” throughout the Discussion section to ensure terminological consistency with the mediation model tested in the study.

5. This sentence was revised for clarity. It now explicitly refers to our findings.

6. The "Implications and Limitations" section was rewritten and expanded. It now discusses:

- The self-reported nature of ADHD diagnosis and potential misclassification bias,

- Self-selection bias related to online recruitment through ADHD-focused groups,

- Limited generalizability due to sample characteristics,

- Lack of control in the online and cross-sectional design,

- Uncontrolled effects of treatment status and psychiatric comorbidities.

These modifications address all aspects raised by the reviewer and align the discussion with best reporting practices.

Additional editorial changes:

a) Reference numbering was adjusted throughout the manuscript and reference list due to removed or condensed citations following the reduction of the Introduction.

b) Minor stylistic corrections were introduced for consistency in terminology.

---

## [Decision Letter · Decision Letter 2]

15 Dec 2025

The mediation effect of general self-efficacy in relation to procrastination and sense of coherence among adults with attention deficit hyperactivity disorder

PONE-D-25-26692R2

Dear Dr. Rodzeń,

We’re pleased to inform you that your manuscript has been judged scientifically suitable for publication and will be formally accepted for publication once it meets all outstanding technical requirements.

Kind regards,

Nicholas Aderinto Oluwaseyi

Academic Editor

PLOS One

Additional Editor Comments (optional):

Reviewers' comments:

Reviewer's Responses to Questions

**Comments to the Author**

Reviewer #1: (No Response)

2. Is the manuscript technically sound, and do the data support the conclusions?

Reviewer #1: (No Response)

3. Has the statistical analysis been performed appropriately and rigorously?

Reviewer #1: (No Response)

4. Have the authors made all data underlying the findings in their manuscript fully available?

Reviewer #1: (No Response)

5. Is the manuscript presented in an intelligible fashion and written in standard English?

Reviewer #1: (No Response)

Reviewer #1: (No Response)

**Do you want your identity to be public for this peer review?** For information about this choice, including consent withdrawal, please see our Privacy Policy

Reviewer #1: No

---

## [Editor Report · Acceptance letter]

PONE-D-25-26692R2

PLOS One

Dear Dr. Rodzeń,

I'm pleased to inform you that your manuscript has been deemed suitable for publication in PLOS One. Congratulations! Your manuscript is now being handed over to our production team.

Kind regards,

on behalf of

Dr. Nicholas Aderinto Oluwaseyi

Academic Editor

PLOS One